# WGREC: WEAKLY SUPERVISED GENERALIZED REFERRING EXPRESSION COMPREHENSION EMPOWERED BY LARGE LANGUAGE MODEL

## ABSTRACT

Weakly Supervised Referring Expression Comprehension (WREC) aims to locate the target object described by a given expression using weak supervision signals, such as image-text pairs. Existing WREC methods typically assume that for every expression, there is always a corresponding object in the image or each frame of a video, ignoring scenarios where multiple objects or no objects match the expression. Additionally, current WREC methods primarily rely on contrastive learning, using numerous positive and negative pairs to construct the loss. This approach has drawbacks: it incurs high computational and memory costs, reduces training efficiency, and is highly sensitive to pair selection, which can lead to unstable convergence or overfitting to specific pairs. In this paper, we introduce a new task, Weakly Supervised Generalized Referring Expression Comprehension (WGREC), which extends traditional WREC to handle more realistic and complex scenarios. To address this task, we design a novel graph-based knowledge distillation network (GKDN) guided by a large language model (LLM). By using the LLM, we obtain two types of information: (1) descriptions of object candidates and their relationships, and (2) pseudo-target positions for single or multiple objects mentioned in the expression. This information helps our network build attention graphs that model the link between objects and the expression while filtering out irrelevant candidates. Finally, a concise objective function is designed, leveraging predictions, expressions, and pseudo target positions, to distill the capabilities of the LLM into our network. Extensive experiments on gRefCOCO, RefCOCO, RefCOCO+, and RefCOCOg datasets demonstrate that our method achieves state-of-the-art (SoTA) performance, highlighting the effectiveness of our approach and its potential to advance the field of WGREC.

## 1 INTRODUCTION

Vision-language representation learning is a fast-growing field in computer vision, supporting tasks like referring expression comprehension (REC) Hamilton et al. (2024); Zhang et al. (2024), generation (REG) Sun et al. (2023a), segmentation (RES) Liang et al. (2022), visual question answering Dancette et al. (2023), image captioning Luo et al. (2023), and scene understanding Peng et al. (2023a).

Among these tasks, weakly supervised referring expression comprehension (WREC) helps connect visual and textual information by identifying objects described in natural language using limited supervision. Unlike fully supervised methods, WREC learns from image-expression pairs without access to exact object locations, requiring the model to infer the link between text and image during training.

While WREC methods have made strong progress, they still face two key challenges. First, most assume each expression refers to exactly one object in the image, which doesn't account for more complex cases where multiple or no objects match the expression. This limits their use in real-world applications. Second, many rely heavily on contrastive learning and complex loss functions, which can hurt generalization. Although recent work Liu et al. (2023a) introduced a broader task called generalized referring expression segmentation to handle more realistic situations, extending WREC

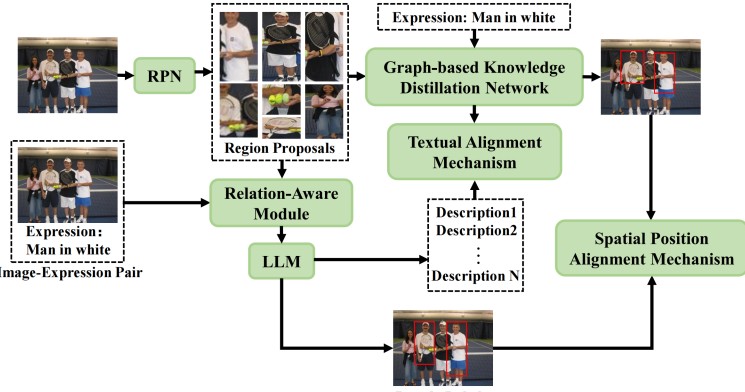

Figure 1: Our method operates differently during the training and inference phases. During training, the LLM enhances the learning process by generating descriptions and pseudo labels, enabling the GKDN to acquire capabilities from the LLM. During inference, only image-expression pairs are input into the GKDN for prediction, ensuring efficient and independent deployment.

to these cases is still unexplored. Also, the method in Liu et al. (2023a) may include irrelevant image regions when selecting object candidates, adding noise that can weaken performance.

Based on the aforementioned issues, a natural question arises: Can traditional WREC be extended to handle more realistic and complex scenarios, and what would be an appropriate method to address this challenge? In this paper, we introduce a new task, termed Weakly Supervised Generalized Referring Expression Comprehension (WGREC), and propose a novel baseline network, referred to as the Graph-Based Knowledge Distillation Network, as illustrated in Fig. 1, to tackle this task. WGREC aims to locate all objects described by the expression in complex scenarios. Moreover, unlike traditional REC tasks, which return the object closest to the expression when no match is found, WGREC will return no object if none of the objects in the image correspond to the description.

During training, we use a large language model (LLM) as the teacher model and our designed model as the student model. In the teacher model, the image is first processed by a Region Proposal Network (RPN) to generate high-confidence region proposals. These proposals, along with the image and the referring expression, are fed into the LLM to generate descriptions and predict pseudo labels for each region proposal. In the student model, the region proposals are passed through a description generator and classifier to produce corresponding descriptions and labels, which help distinguish between proposals more precisely. These region proposals, along with their labels and descriptions, are then input into our dynamic multi-graph attention module to model relationships between the proposals. The output of this module, combined with the referring expression, is used in our evaluation strategy to generate the final prediction. The descriptions produced by our method are aligned with those from the LLM, improving the model's ability to replicate the LLM's performance. During inference, only our proposed method is used for prediction.

In summary, there are three contributions in this paper.

- To the best of our knowledge, we are the first to extend traditional WREC to more realistic and complex scenarios. Our introduced framework overcomes the limitations of existing methods by enabling reliable multi-object localization, handling incomplete or noisy annotations.

- To the best of our knowledge, we are the first to apply an LLM-based knowledge distillation strategy to the WREC task. This new approach improves performance in complex and realistic settings. By using the power of LLMs, our network becomes more compact and accurately models the relationships between target objects and expressions, leading to better localization and understanding even in difficult cases.

- We introduce a novel evaluation strategy that enables our method to better adapt to the diverse scenarios in GREC, including cases with no target, a single target, or multiple targets.

## 2 RELATED WORKS

### 2.1 WEAKLY SUPERVISED REFERRING EXPRESSION COMPREHENSION (WREC)

Recent state-of-the-art REC methods mostly use transformers and need large, fully annotated datasets for training. Although they perform well, they require a lot of computing power and manual labeling. To address this, WREC methods have been developed, using weaker supervision like image-text pairs to cut down on annotation effort and resource use. Early WREC methods Gpta et al. (2020); Liu et al. (2019a;b; 2021); Sun et al. (2021); Wang et al. (2021); Zhang et al. (2020); Liu et al. (2023b); Jiang et al. (2022); Liu et al. (2023c); Ji et al. (2024); Sun et al. (2023b); Chen et al. (2018) primarily focused on two-stage frameworks. Among these, Liu et al. (2019b; 2023b); Sun et al. (2021; 2023b); Chen et al. (2018); Ji et al. (2024); Jiang et al. (2022); Liu et al. (2023c); Jiang et al. (2022); Liu et al. (2023c) introduce a reconstruction-based strategy that generates descriptions for object proposals in an image. These generated descriptions, along with the input expression and object proposals, are then combined and fed into vision-language interaction modules to localize the target object, guided by contrastive learning techniques. Among, Liu et al. (2021) proposes a coarse-to fine graph-based method to model relationships between object proposals during their interaction with the expression, improving localization performance. Wang et al. (2021) introduces a knowledge distillation strategy, aiming to extract the capabilities of an object detector to enhance performance. In contrast, existing one-stage WREC methods Jin et al. (2023); Luo et al. (2025); Zhao et al. (2018) employ anchor-text matching combined with contrastive learning for prediction.

However, existing WREC methods face two key challenges, as most assume each expression refers to exactly one object—ignoring cases with multiple or no matches—and many rely on complex contrastive learning frameworks that hinder generalization. To address these issues, we introduce a new task, Weakly Supervised Generalized Referring Expression Comprehension (WGREC), which extends WREC to more complex and realistic scenarios by handling multiple objects, modeling their relationships, and learning from limited labeled data. Furthermore, we propose a Graph-based Knowledge Distillation Network (GKDN) tailored for WGREC, which alleviates the limitations of existing WREC approaches, achieves state-of-the-art performance, and generalizes effectively to this broader task.

### 2.2 VISION-LANGUAGE LARGE MULTIMODAL MODELS FOR REC

Leveraging the remarkable generalization capabilities of LLMs, recent studies have expanded their application to multi-modal domains by aligning visual inputs with LLMs. Early works, such as VisualGPT Chen et al. (2022) and Frozen Tsimpoukelli et al. (2021), leveraged pre-trained language models to enhance vision-language tasks like image captioning and visual question answering. These foundational efforts laid the groundwork for subsequent advancements in vision-language research, including Flamingo Alayrac et al. (2022) and BLIP-2 Li et al. (2023). In the context of REC, pioneering works Peng et al. (2023b); Chen et al. (2023b;a); Zhang et al. (2025); Guo et al. (2024) focus on integrating the target object to be localized and representing object locations using textual coordinates or coordinate placeholders. This approach equips models with initial capabilities for target object prediction. Subsequent research Wang et al. (2024) further improves REC performance by incorporating data from diverse visual tasks and introducing more flexible referencing methods. Notably, Griffon Zhan et al. (2025) and Griffon-v2 Zhan et al. (2024) unify localization tasks of varying granularities through next-token prediction, enabling Large Multi-modal Models (LMMs) to handle complex visual tasks included REC. Additionally, studies Jiao et al. (2024); Wang et al. (2023); Zhao et al. (2023); Ma et al. (2025) enhance the positional perception abilities of models in visual tasks by integrating visual expert models or specialized decoding structures. These methods provide valuable insights into enabling LMMs to tackle both visual and vision-language tasks effectively.

To the best of our knowledge, LLMs have not yet been applied to the WREC task. In this paper, we propose a method guided by knowledge distilled from LLMs. By leveraging this knowledge, our approach achieves a more compact model size while maintaining competitive performance on both WREC and WGREC tasks. Additionally, we introduce a new evaluation strategy that better captures the flexible and complex nature of GREC scenarios.

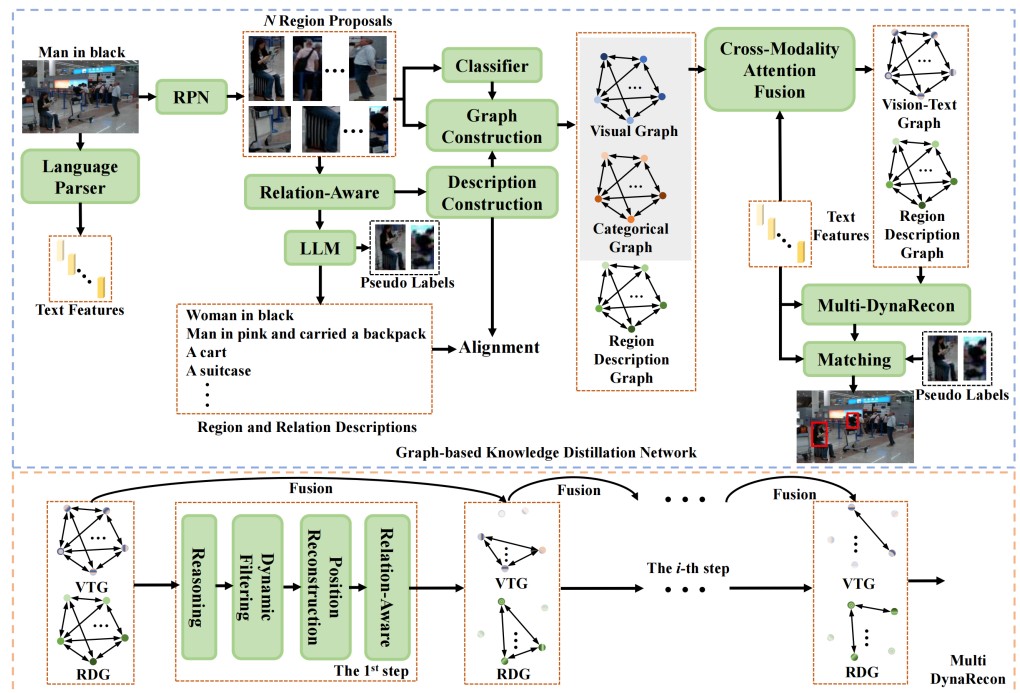

Figure 2: The object proposals and input expression are processed by a large language model (LLM) and a language parser to get region descriptions and text features. The LLM's descriptions serve as pseudo-labels to help the description module generate better descriptions. Each proposal, along with its category and description, is used to build three graphs: a visual graph, a category graph, and a description graph. The visual and category graphs are combined using cross-modal attention to create a vision-text graph. This graph is then merged with the description graph and passed through the multi-DynaRecon module to find the regions most relevant to the expression. VTG and RDG denote the vision-text graph and the region-description graph, respectively.

## 3 METHODOLOGY

Figure 2 illustrates our proposed framework, detailing both the training and sampling phases. The Graph-Based Knowledge Distillation Network (GKDN) has four main parts: a Relation-Aware Module that captures relationships between object proposals, a Graph Construction Module that builds three types of graphs, a Cross-Modality Attention Module that combines information from different graphs, and a Multi-DynaRecon Module that gradually refines the graph nodes most relevant to the input expression.

### 3.1 PRELIMINARY

Given an image $I$ and an expression $Q$ composed of $L$ words, we first employ a RPN network to generate $K$ object proposals, denoted as $\boldsymbol{O} = \left\{o_i \in \mathbb{R}^{D_o}\right\}_{i=1}^{K}$. These proposals are then processed by a classifier to obtain their corresponding categorical labels $\boldsymbol{C} = \left\{c_i \in \mathbb{R}^{D_c}\right\}_{i=1}^{K}$. For the expression $Q$, we use BERT to embed each token as well as the entire expression. The resulting token embeddings are represented $\left\{\gamma_i \in \mathbb{R}^{d_\gamma}\right\}_{i=1}^{L}$, while the full expression embedding is denoted as $q$.

**Relation-Aware Module.** Previous reconstruction-based WREC methods reconstruct each object proposal's expression independently, ignoring relationships between proposals. To address this limitation, we propose a relation-aware module that generates new regions by combining object proposal positions. These reconstructed regions are then processed by the description construction module to recover relationship descriptions. Specifically, let the positions of the $i$-th and $j$-th objects be $[left_i, top_i, right_i, bottom_i]$ and $[left_j, top_j, right_j, bottom_j]$, respectively. A new region $pos_{ij}$ is constructed as: $pos_{ij} =$

$[max(left_i, left_j), max(top_i, top_j), max(right_i, right_j), max(bottom_i, right_j)]$. Here, content outside the overlapping area of proposals $i$ and $j$ is masked. The resulting region is then fed into the description construction module for description generation.

**LLMs.** To effectively leverage the knowledge from the LLMs, we first input the object proposals directly into the LLMs to generate descriptions for each proposal. The resulting object descriptions are denoted as $\boldsymbol{P} = \left\{ p_i \in \mathbb{R}^{D_p} \right\}_{i=1}^K$. Next, we combine every pair of object proposals with the entire image and feed them into the LLM to obtain corresponding relationship descriptions, denoted as $\boldsymbol{R} = \left\{ r_{ij} \in \mathbb{R}^{D_r} \right\}_{i,j=1}^K$.

## 3.2 GRAPH CONSTRUCTION

**Visual, Categorical and Description Graph.** After obtaining the object proposals along with their corresponding categorical labels and descriptions, we construct three graphs: a visual graph $G_a$, a categorical graph $G_c$ and a description graph $G_p$. For the visual graph $G_a = (V_a, E_a)$, the node features are denoted as $V_a = \left\{ v_i^a \in \mathbb{R}^{d_a} \right\}_{i=1}^k$, and the edge features are $E = \left\{ e_{ij}^a \in \mathbb{R}^{d_a} \right\}_{i,j=1}^k$. The $i$-th node feature of $G_a$ is computed as:

$$v_i^a = \mathbf{W}_{vis}\left([o_i; \mu_i]\right) + b_a, \tag{1}$$

where $\mathbf{W}_{vis}$ is a trainable weight matrix. $b_a$ is a trainable bias vector. $o_i$ denotes the visual features of the $i$-th object proposal. $\mu_i = \mathbf{W}_\mu [x_i, y_i, w_i, h_i, w_i h_i]$ encodes the spatial features of the $i$-th proposal. Here, $(x_i, y_i)$ are the normalized 2D coordinates of its center, while $w_i$, $h_i$, and $w_i h_i$ represent the normalized width, height, and area, respectively. $\mathbf{W}_\mu$ is a trainable parameters matrix. The edge features between the $i$-th and $j$-th node of $G_a$ is denoted as

$$e_{ij}^a = \mathbf{W}_{edge}\left(\left[v_i^a; v_j^a; \mu_i; \mu_j\right]\right), \tag{2}$$

where $\mathbf{W}_{edge}$ is a trainable weight matrix.

For the categorical graph $G_c$, the node and edge weights are computed similarly, except that all visual features (e.g., $o_i$) are replaced by their corresponding textual features.

For the description graph $G_p = (V_p, E_p)$, the node features $V_p = \left\{ v_i^p \in \mathbb{R}^{d_p} \right\}_{i=1}^k$ and edge features $E = \left\{ e_{ij}^p \in \mathbb{R}^{d_p} \right\}_{i,j=1}^k$ are generated by the description construction module, which encodes linguistic information from object descriptions $\boldsymbol{P}$ and their relationships $\boldsymbol{R}$.

**Cross-Modality Attention Fusion Module.** To improve object proposal representation and strengthen the graph nodes relevant to the referring expression, we extract each node from both the visual graph $G_a$ and categorical graph $G_c$ to compute its similarity with the expression. Formally, for the $i$-th node of $G_a$, the similarity scores are calculated as:

$$s_i^a = \mathbf{W}_a[\tanh(v_i^a + \mathbf{W}_q q)], \tag{3}$$

where $\mathbf{W}_a$ and $\mathbf{W}_q$ denote trainable parameter matrices. The similarity scores between the edges of $G_a$ and the expression are similarly computed as

$$\tau_{ij}^a = \mathbf{W}_{edge,a}[\tanh(e_{ij}^a + \mathbf{W}_q q)], \tag{4}$$

where $\mathbf{W}_{edge,a}$ denote trainable parameter matrix. The computation of node and edge scores for categorical graphs is similar to that of visual graphs, except the label 'a' is replaced with 'c'.

Then, we combine and fuse the nodes and edges from both graphs with their corresponding similarity scores and the expression to construct a vision-text graph $G_{ac}$. The $i$-th node of $G_{ac}$ is computed as

$$\begin{aligned} v_i^{ac} &= s_i^a v_i^a + s_i^c v_i^c, \\ v_i^{ac} &= \mathbf{W}_{e2}\left(tanh\left(\mathbf{W}_{e1}\left[v_i^{ac}; q\right]\right)\right), \end{aligned} \tag{5}$$

where $\mathbf{W}_{e1}$ and $\mathbf{W}_{e2}$ denote trainable parameter matrices. Furthermore, the node weight of the $i$-th node of $G_{ac}$ is computed as

$$\begin{aligned} s_i^{ac} &= s_i^a + s_i^c, \\ w_i^{ac} &= \frac{exp\left(s_i^{ac}\right)}{\sum_{n=1}^K exp\left(s_n^{ac}\right)}. \end{aligned} \tag{6}$$

Similarly, the edge weight of the $G_{ac}$ is computed as

$$\tau_{ij}^{ac} = \tau_{ij}^{a} + \tau_{ij}^{c},$$
$$w_{ij}^{ac} = \frac{exp\left(\tau_{ij}^{ac}\right)}{\sum_{n=1}^{K} exp\left(\tau_{in}^{ac}\right)}. \tag{7}$$

For the description graph $G_p$, the node and edge weights are computed similarly to the visual graph, denoted as $\{w_i^p\}_{i=1}^{K}$ and $\{w_{ij}^p\}_{i,j=1}^{K}$.

### 3.3 MULTI-DYNARECON

Using the input expression as guidance, we apply a one-step reasoning method from Ke et al. (2025) to both the vision-text and description graphs. This reduces the impact of unrelated nodes. We then combine the refined graphs to improve the quality of the descriptions.

Specifically, let the nodes of the vision-text graph $G_{ac}(t)$ and the description graph $G_p(t)$ at $t$ reasoning step ($t \geq 1$) reasoning step be denoted as $\{v_i^{ac}(t)\}_{i=1}^{K}$ and $\{v_i^{p}(t)\}_{i=1}^{K}$, respectively. The nodes in the vision-text graph that are identified as strongly related to the expression—based on the strategy in Ke et al. (2025)—are passed through an MLP module to generate $K_t$ four-dimensional positional embeddings, where $K_t$ is the number of such labeled nodes at step $t$. In other words, each labeled node (high related to the expression) produces a 4D positional embedding.

These $K_t$ generated positions, along with the full image, are passed through the relation-aware module and the description construction module to refine both object descriptions and their corresponding relationship descriptions. Finally, we update $G_{ac}(t)$ and $G_p(t)$ by fusing the refined descriptions and vision-text graph nodes at step $t$ with their counterparts from step $t-1$. In this process, the edge features of $G_{ac}(t)$ are computed as:

$$e_{ij}^{ac}(t) = \mathbf{W}_{ac,t}\left(\left[v_i^{ac}(t); v_j^{ac}(t); q\right]\right), \tag{8}$$

where $\mathbf{W}_{ac,t}$ is the trainable parameter matrix. The node and edge weights of $G_{ac}(t)$ and $G_p(t)$ are computed similarly to $G_a$ (i.e., following Eq. 3 and 4).

### 3.4 MATCHING

Previous method Liu et al. (2023a) used a two-stage evaluation strategy for GREC during loss function design. It first used a classifier to check if the target object mentioned in the expression exists in the image. If so, it then searched for related object proposals using a threshold. This process reduced localization efficiency. To improve this, after reasoning over $T$, we design the matching loss with a novel evaluation strategy that uses only one threshold for object searching. For each example, we calculate similarity scores between the final aggregated node features from both graphs ($G_{ac}(T)$ and $G_p(T)$) and the textual features of the entire expression $q$, as follow:

$$\xi_i^* = \left\langle \frac{\mathbf{W}_M v_i^*(T)}{\|\mathbf{W}_M v_i^*(T)\|}, \frac{\mathbf{W}_q q}{\|\mathbf{W}_q q\|} \right\rangle * \in \{ac, p\}, \tag{9}$$

where $\mathbf{W}_M$ and $\mathbf{W}_q$ represent trainable parameter matrices.

For samples with label $r = 1$ (indicating one or more objects match the expression), $Y$ denote the number of pseudo ground-truth (PGT) objects generated by the LLM. We compute the matching loss using the similarity scores $\xi_{i,gt}^*$ of PGT nodes in both graphs as follows:

$$P_{gt} = \frac{\sum_{i=1}^{Y} \frac{exp\left(\xi_{i,pgt}^{ac} + \xi_{i,pgt}^{p}\right)}{\sum_{j=1}^{K} exp\left(\xi_j^{ac} + \xi_j^{p}\right)}}{Y}, \tag{10}$$
$$\mathcal{L}_s = -\log\left(P_{\text{pgt}}\right).$$

To enhance bounding box accuracy, we first concatenate the expression features with the ground-truth node features from both graphs. These combined features are then used to refine the target

Table 1: Comparison between our method and other approaches on the gRefCOCO dataset. $\mathrm{MiniGPT-v2}^{\dagger}$ and $\mathrm{MiniGPT-v2}^{*}$ represent MiniGPT-v2 denote MiniGPT-v2 evaluated using the strategy from Liu et al. (2023a) and our evaluation strategy, respectively. GKDN w/ F and GKDN w/ W refer to our GKDN model under fully supervised and weakly supervised settings, respectively. The best results are shown in bold, and weakly supervised results are shown in blue.

| Methods | Visual Encoder | val | | testA | | testB | |
|---|---|---|---|---|---|---|---|
| | | Pr@(F1=1, IoU≥0.5) | N-acc | Pr@(F1=1, IoU≥0.5) | N-acc | Pr@(F1=1, IoU≥0.5) | N-acc |
| MCN | DarkNet-53 | 28.0 | 30.6 | 32.3 | 32.0 | 26.8 | 30.3 |
| VLT | DarkNet-53 | 36.6 | 35.2 | 40.2 | 34.1 | 30.2 | 32.5 |
| MDETR | ResNet-101 | 42.7 | 36.3 | 50.0 | 34.5 | 36.5 | 31.0 |
| UNINEXT | ResNet-50 | 58.2 | 50.6 | 46.4 | 49.3 | 42.9 | 48.2 |
| $\mathrm{MiniGPT-v2}^{\dagger}$ | ViT | **61.7** | **52.2** | 50.6 | **51.5** | 45.4 | **50.7** |
| $\mathrm{MiniGPT-v2}^{*}$ | ViT | 60.8 | 51.7 | **51.0** | 50.8 | **45.8** | 50.3 |
| GKDN w/ F | ResNet-101 | 45.2 | 44.6 | 49.7 | 38.3 | 38.5 | 33.7 |
| GKDN w/ W | ResNet-101 | 32.7 | 31.5 | 35.3 | 23.4 | 27.2 | 19.8 |

Table 2: Comparison between our method and other WREC approaches on the RefCOCO, /+/g datasets.

| Methods | Visual Encoder | RefCOCO | | | RefCOCO+ | | | RefCOCOg |
|---|---|---|---|---|---|---|---|---|
| | | val | testA | testB | val | testA | testB | val-g |
| VC | VGG16 | - | 32.68 | 27.22 | - | 34.68 | 28.10 | 29.65 |
| IGN | ResNet101 | 34.78 | - | - | - | 36.91 | 36.91 | 35.46 |
| ARN | ResNet101 | 32.17 | 35.25 | 30.28 | 32.78 | 34.35 | 32.13 | 33.09 |
| DTWREG | ResNet101 | 38.35 | 39.51 | 37.01 | 38.19 | 39.91 | 37.09 | 42.54 |
| APL | DarkNet-53 | 64.51 | 61.91 | **63.57** | 42.70 | 42.84 | 39.80 | 50.22 |
| RefCLIP | DarkNet-53 | 60.36 | 58.58 | 57.13 | 40.39 | 40.45 | 38.86 | 47.87 |
| GKDN | ResNet101 | **67.84** | **65.84** | 62.18 | **47.65** | **46.28** | 42.36 | **53.34** |

object's bounding box through additional fully-connected layers, supervised by a smooth $\mathcal{L}1$ loss ($\mathbf{SmoothL}1(\cdot, \cdot)$) between predicted and ground-truth boxes:

$$b_{\mathrm{i,pred}} = \mathbf{MLP}\left(\left[v_{\mathrm{i,pgt}}^{ac}(T); v_{\mathrm{i,pgt}}^{p}(T)\right]; q\right),$$
$$\mathcal{L}_{\mathrm{reg}} = \frac{\sum_{i=1}^{Y} \mathbf{SmoothL}_1(b_{\mathrm{i,pred}}, b_{\mathrm{i,pgt}})}{Y}, \tag{11}$$

where $b_{\mathrm{i,pred}}$ denotes the $i$-th predicted 4D bounding box vector (output by an MLP and $b_{\mathrm{i,pgt}}$ represents the $i$-th pseudo ground-truth box.

For samples with label $r = 0$ (indicating no target object matches the expression), the loss function is defined as:

$$\mathcal{L}_{\mathrm{neg}} = \frac{\sum_{i=1}^{K} \xi_i^{ac} + \xi_i^{p}}{2K}, \tag{12}$$

Thus, the final loss of our method is defined as:

$$\mathcal{L} = \mathcal{L}_{\mathrm{s}} + \mathcal{L}_{\mathrm{reg}} + \mathcal{L}_{\mathrm{neg}}, \tag{13}$$

During inference, we compute node-expression similarity scores between both graphs' features ($G_{ac}(T)$, $G_p(T)$) and $q$. If all $K$ scores ($\frac{\xi_i^{ac} + \xi_i^{p}}{2}$) fall below threshold, no target exists. Otherwise, we consider the prediction correct if: (1) the number of matching nodes $\geq$ ground-truth boxes, and (2) all corresponding predicted boxes have IoU $\geq 0.5$ with ground-truth boxes (keeping only the highest-IoU prediction when multiple boxes match one ground-truth).

## 4 EXPERIMENTAL RESULTS

In this section, we present extensive evaluation results of the proposed method on three challenging REC benchmarks—RefCOCO Kazemzadeh et al. (2014), RefCOCO+ Kazemzadeh et al. (2014), and RefCOCOg Mao et al. (2016)—as well as one challenging GREC benchmark: gRefCOCO Liu et al. (2023a). RefCOCO and RefCOCO+ contain relatively short expressions, with average lengths of 3.61 and 3.65 words, respectively. In contrast, RefCOCOg and Ref-reasoning feature longer and more complex expressions, averaging 8.4 and 8.5 words, respectively. Due to limited space, we provide detailed information about these datasets and implementation details in the supplementary materials.

Table 3: Ablation study of our method with different LLMs on the gRefCOCO dataset. GKDN w/ M and GKDN w/ G represent GKDN using MiniGPT-v2 and Grounding DINO as the teacher model, respectively.

| Methods | Visual Encoder | val | | testA | | testB | |
|---|---|---|---|---|---|---|---|
| | | Pr@(F1=1, IoU≥0.5) | N-acc | Pr@(F1=1, IoU≥0.5) | N-acc | Pr@(F1=1, IoU≥0.5) | N-acc |
| MiniGPT-v2 (untrained) | ViT | 43.65 | 39.72 | 40.14 | 39.67 | 31.42 | 38.70 |
| MiniGPT-v2 | ViT | 60.83 | 51.72 | 51.03 | 50.85 | 45.83 | 50.37 |
| Grounding DINO | ViT | 42.84 | 41.35 | 41.54 | 41.62 | 32.38 | 40.26 |
| GKDN w/ M (untrained) | ResNet-101 | 14.34 | 25.83 | 23.25 | 14.64 | 17.82 | 15.62 |
| GKDN w/ L | ResNet-101 | 15.44 | 25.63 | 24.74 | 15.52 | 19.05 | 14.93 |
| GKDN w/ M | ResNet-101 | 32.75 | 31.54 | 35.32 | 23.43 | 27.24 | 19.85 |

Table 4: Ablation studies of our method using different numbers of graphs. V, C, and P stand for the visual, categorical, and description graph.

| V | V+C | V+C+P | Val | | TestA | | TestB | |
|---|---|---|---|---|---|---|---|---|
| | | | Pr@(F1=1, IoU≥0.5) | N-acc | Pr@(F1=1, IoU≥0.5) | N-acc | Pr@(F1=1, IoU≥0.5) | N-acc |
| ✓ | | | 30.83 | 30.07 | 33.52 | 21.75 | 25.44 | 18.73 |
| ✓ | ✓ | | 31.33 | 30.26 | 33.74 | 22.68 | 25.94 | 18.75 |
| ✓ | ✓ | ✓ | 32.75 | 31.54 | 35.32 | 23.43 | 27.24 | 19.85 |

## 4.1 EVALUATION RESULTS

We first present the evaluation results of our method (GKND) and other baseline approaches on the gRefCOCO dataset. To adapt MiniGPT-v2 to this task, we integrate it with our proposed evaluation evaluation and fine-tune it on the gRefCOCO dataset. As shown in the second-to-last part of Table 1, MiniGPT-v2 achieves the best performance. Additionally, when applying the two-stage evaluation strategy from Liu et al. (2023a) to MiniGPT-v2, we observe that its performance under both their strategy and ours is very similar. However, our evaluation strategy relies solely on a threshold-based approach, without the need for a two-stage strategy, making it more efficient than the method proposed in Liu et al. (2023a).

In the bottom section of Table 1, under the fully supervised setting, GKDN outperforms most SoTA methods, with the exception of MiniGPT-v2 and UNINEXT, which rely on large pre-trained language models and require significantly more computational resources. Under the weakly supervised setting, guided by MiniGPT-v2, our method still surpasses several SoTA methods. These results demonstrate that GKND is competitive under full supervision on gRefCOCO and has strong potential for WGREC.

We further compare our method with WREC methods on the RefCOCO /+/g datasets. The results in Table 2 show that, on average, our method outperforms the current SoTA WREC method (APL) by about 2.84%. Taken together, the results from Tables 1 and 2 confirm that our method achieves strong performance in both generalized referring expression comprehension (GREC) and WREC, especially with the assistance of large language models (LLMs). Although our performance in WGREC is not yet remarkable, as the first work in this area, it demonstrates significant promise and establishes a meaningful foundation for future research.

## 4.2 ABLATION STUDIES

We begin by evaluating the performance of our method using different LLMs. As shown in Table 3, without training, both MiniGPT-v2 and Grounding DINO (11B) perform noticeably worse than the trained version of MiniGPT-v2. Additionally, our method performs slightly better when using Grounding DINO as the teacher model compared to the untrained MiniGPT-v2. Aforementioned results suggest that current smaller-scale LLMs still hold significant potential for the GREC task. Using a larger model such as Grounding DINO may further improve GKDN's performance.

Later, we analyze the effectiveness of the visual graph, categorical graph, and description graph in our network. The results presented in Table 4 show that the performance of GKDN gradually improves when the categorical and description graphs are combined. Specifically, when the categorical graph is fused with the visual graph, the performance of our method improves by an average of 0.39%. However, when the description graph is considered, the performance improves by an average of 1.23%. These results show that both the categorical and description graphs provide complementary and valuable information, helping GKDN achieve better performance.

Table 5: Ablation study of multi-DynaRecon on the gRefCOCO dataset. GKDN w/o R and GKDN w/o F indicate GKDN without the description strategy and graph fusion strategy, respectively.

| Methods | Visual Encoder | val | | testA | | testB | |
|---|---|---|---|---|---|---|---|
| | | Pr@(F1=1, IoU$\geq$0.5) | N-acc | Pr@(F1=1, IoU$\geq$0.5) | N-acc | Pr@(F1=1, IoU$\geq$0.5) | N-acc |
| GKDN w/o R | ResNet-101 | 31.25 | 30.58 | 34.66 | 22.94 | 26.75 | 18.72 |
| GKDN w/o F | ResNet-101 | 31.68 | 30.74 | 34.83 | 23.07 | 26.87 | 19.45 |
| GKDN | ResNet-101 | 32.75 | 31.54 | 35.32 | 23.43 | 27.24 | 19.85 |

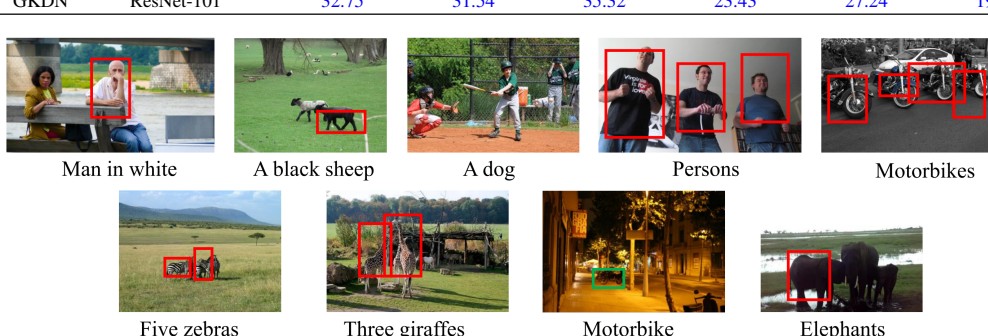

| Man in white | A black sheep | A dog | Persons | Motorbikes |
|---|---|---|---|---|

| Five zebras | Three giraffes | Motorbike | Elephants |
|---|---|---|---|

Figure 3: Visualization of our method on WREC and GWREC tasks. The first two samples in the top row show results on the WREC task, while the remaining samples show results on the GWREC task. Red bounding boxes represent the predicted results, and green bounding boxes indicate the ground truth.

Finally, let's focus on the advantages of our proposed multi-dynaRecon module. The results presented in Table 5 show that removing the description reconstruction and graph fusion strategies leads to a slight decrease in performance. These findings demonstrate that gradually fusing graphs from previous reasoning steps helps accumulate valid information. Furthermore, the combination of the reconstruction operation and the removal of unrelated nodes effectively guides the model to focus on object proposals that are highly relevant to the expression.

### 4.3 VISUALIZATION

"In addition to the quantitative results, we also provide visual examples of our method in Figure 3. The first row shows accurate predictions made by our method, while the second row illustrates failure cases. In the first row, the first two samples present results from the WREC task, and the remaining three are from the GWREC task. It is evident that our method performs well on both tasks. Notably, in the third sample, no objects are related to the 'dog', so the proposed method doesn't output any bounding boxes. In the fifth sample, although there are four motorbikes with identical shapes, our method successfully locates each one correctly. In contrast, the second row highlights cases of incorrect predictions. In the first sample, the target objects are clustered and partially overlapping, resulting in incorrect localization. In the second sample, one 'giraffe' is occluded by the other, preventing the model from accurately locating it. In the last two samples, the model fails to detect all targets due to the dark background.

## 5 CONCLUSION

In this paper, we make a pioneering contribution by introducing a new task—Weakly Supervised Generalized Referring Expression Comprehension (WGREC)—which, to the best of our knowledge, is the first to extend traditional WREC to more realistic and complex scenarios. To address this task, we propose a novel Graph-based Knowledge Distillation Network (GKDN) guided by a Large Language Model (LLM). The LLM is used to generate enriched descriptions for object proposals, their relationships, and pseudo-target positions in both single-object and multi-object scenarios described by the expressions. These enhanced representations help our network construct a series of attention graphs that effectively model the relationships between object candidates and the referring expression while filtering out irrelevant candidates. Furthermore, we introduce a new evaluation strategy that remains compatible with existing REC methods. Extensive experiments on four datasets show that our method achieves state-of-the-art performance on the WREC task. Although our method's performance on WGREC is not yet outstanding, this work serves as a foundational step, offering valuable insights for future research in this direction.

**Ethics Statement.** This work does not involve human subjects, personal or sensitive data, or experiments that pose risks to people, society, or the environment. All datasets used are publicly available and contain no personally identifiable information. We confirm that this research complies with the ICLR Code of Ethics. We used GPT-4 solely for language polishing and improving readability. All technical content, including ideas, methodology, experiments, and analysis, was designed and written by the authors, who take full responsibility for the final content.

**Reproducibility Statement.** We have taken several steps to ensure the reproducibility of our work. All datasets used in this study are publicly available and referenced in the main text. Detailed descriptions of the model architecture, training procedure, and hyperparameters are provided in the appendix. Additional experimental results and ablation studies are included in the supplementary material. To further support reproducibility, we will release the source code and instructions for reproducing the experiments as anonymous supplementary material at the time of submission.

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

## A APPENDIX

### A.1 IMPLEMENTATION DETAILS

In this work, we use the region proposal network (RPN) from Chen et al. (2021) to generate object proposals, and adopt the classifier from Anderson et al. (2018) to classify them. The RPN is based on a ResNet-101 backbone and the LLM model we used as a teacher model is Grounding DINO Liu et al. (2025) and MiniGPT-v2 Chen et al. (2023a). To prevent data leakage during training, we remove overlapping image categories between the COCO training set and the four datasets used in our

Table 6: Ablation studies of our method in different reasonig steps.

| 2 | 3 | 4 | Val | | TestA | | TestB | |
|---|---|---|---|---|---|---|---|---|
| | | | Pr@(F1=1, IoU$\geq$0.5) | N-acc | Pr@(F1=1, IoU$\geq$0.5) | N-acc | Pr@(F1=1, IoU$\geq$0.5) | N-acc |
| ✓ | | | 32.08 | 31.35 | 34.44 | 22.93 | 26.84 | 19.56 |
| | ✓ | | 32.75 | 31.54 | 35.32 | 23.43 | 27.24 | 19.85 |
| | | ✓ | 31.25 | 30.83 | 34.16 | 21.95 | 26.08 | 18.27 |

experiments. Furthermore, the evaluation setting for our method on the GREC dataset gRefCOCO follows the protocol in He et al. (2023). The reasoning step $T$ is set to 3. A threshold of 0.7 is used to determine whether an object matches the expression—specifically, if the score of a node in the graph exceeds 0.7, the node is considered relevant to the expression. A prediction is deemed correct if the Intersection-over-Union (IoU) between the predicted bounding boxes of the selected nodes and the ground-truth bounding boxes exceeds 0.5.

## A.2 DATASETS

- **RefCOCO**, **RefCOCO+**, and **RefCOCOg:** The RefCOCO and RefCOCO+ datasets contain 142,210 and 141,564 expressions referring to 50,000 and 49,856 objects across 19,994 and 19,992 images, respectively. These expressions were collected through an interactive game. Kazemzadeh et al. Kazemzadeh et al. (2014) divided RefCOCO into training, validation, testA, and testB sets, with 120,624, 10,834, 5,657, and 5,095 expression-object pairs, respectively. TestA focuses on images with multiple people, while testB contains images with multiple non-human objects. RefCOCO+ uses the same split, with 120,191, 10,758, 5,726, and 4,889 pairs for training, validation, testA, and testB, respectively. Unlike RefCOCO, RefCOCO+ excludes expressions based on absolute location. RefCOCOg, in contrast, was collected in a non-interactive setting and includes 95,010 longer expressions referring to 49,822 objects in 25,799 images. It is split into 80,512 for training, 4,896 for validation, and 9,602 for testing. All three datasets are derived from MSCOCOPont-Tuset & Van Gool (2015) and span 80 object categories.

- **gRefCOCO:** The dataset includes 278,232 expressions, covering 60,287 unique instances across 19,994 images. Among these, 80,022 are multi-target expressions and 32,202 are no-target expressions. All target instances are annotated with both masks and bounding boxes. Some single-target expressions are inherited from RefCOCO. The annotation process follows the ReferIt protocol Kazemzadeh et al. (2014) to ensure high quality, and the data split aligns with the UNC partition of RefCOCO Kazemzadeh et al. (2014).

## A.3 MORE ANALYSIS OF OUR METHOD

In this section, we investigate the effect of the number of reasoning steps on our method. As shown in Table 6, when the number of reasoning steps is fewer than 3, the performance of our method improves with each additional step. However, when the number of steps reaches 4, performance begins to decline. This suggests that exceeding 3 reasoning steps may lead to overfitting.

