# OpenReview forum: "WGREC: Weakly Supervised Generalized Referring Expression Comprehension Empowered by Large Language Model"
_ICLR.cc/2026/Conference — ICLR 2026 Conference Withdrawn Submission_

### Official Review · Reviewer_11sh · 2025-10-19

**Soundness:** 3
**Presentation:** 3
**Contribution:** 3
**Rating:** 6
**Confidence:** 3

**Summary:**

This paper frames weakly supervised generalized referring expression comprehension, where an expression may match zero, one, or many objects and proposes GKDN, a student model trained with an LLM teacher that, given detector proposals plus the image and expression, generates object/relationship descriptions and pseudo target boxes for distillation; at inference, only the compact student runs (no VLM/LLM), using a single-threshold scorer to decide cardinality. The student builds visual, categorical, and description graphs, fuses them with cross-modal attention, and learns via a concise objective combining matching, box regression, and negative-case penalties. Experiments on gRefCOCO and RefCOCO/+/g show competitive WGREC performance and an average +2.84% over APL on WREC, while ablations indicate that teacher quality materially affects the distilled student and that both categorical and description graphs add complementary gains.

**Strengths:**

1. Graph-centric reasoning that reflects language structure. Three graphs (visual/category/description) + one-step reasoning per Ke et al. are fused and iterated; ablations show each graph contributes (Table 4) and that reasoning/fusion helps (Table 5).

2. Practical evaluation tweak. A single-threshold matcher replaces the two-stage existence-check in prior GREC work, simplifying training/eval without hurting MiniGPT-v2 performance.

3. Solid performance from GKDN compared with previous methods.

4. The paper is well-written and easy to follow

**Weaknesses:**

1. Teacher quality matters and might be more important that proposed  Graph-centric reasoning modules. Table 3 shows strong dependence on the chosen teacher (MiniGPT-v2 trained > untrained; Grounding DINO in between). Distillation quality bounds student performance.

2. Compute/latency not reported. Graph construction + pairwise relations + reasoning can be costly; O(K²) edges suggest practical limits—no fps/memory table yet.

**Questions:**

1. Could you report GKDN trained with two different teachers (MiniGPT-v2 vs. Grounding DINO) under identical proposal prompts to isolate teacher effects on WGREC? (Table 3 hints at this but mixes conditions.)

2. What are K (proposals per image), edge counts, and end-to-end latency/memory for training and inference? A small table would help position GKDN vs APL/RefCLIP.

3. You set T=3 based on Table 6; can you expose an early-exit rule when the top-K stabilizes to save compute at inference?

4. Your matcher hinges on a single score threshold (0.7 in App. A.1); how sensitive are WGREC metrics to this choice?

---

### Official Review · Reviewer_Kho9 · 2025-10-28

**Soundness:** 3
**Presentation:** 3
**Contribution:** 2
**Rating:** 4
**Confidence:** 4

**Summary:**

The paper proposes the Weakly Supervised Generalized Referring Expression Comprehension (WGREC) task, which can cover multi-target and no-target matching scenarios, and explicitly returns "no target" when there is no match. Meanwhile, it designs a Graph-based Knowledge Distillation Network (GKDN), with a Large Language Model (LLM) serving as the teacher model to generate key information. GKDN distills the capabilities of the LLM into the network through its core modules.

**Strengths:**

The proposed method outperforms traditional approaches and establishes the first baseline. The pipeline of the proposed method is clear.

**Weaknesses:**

1. This paper primarily combines several existing works and applies them to a new scenario. In addition, incorporating LLMs through knowledge distillation lacks significant novelty in terms of methodological design.

2. The first and second contributions appear highly similar, and the third contribution is not clearly explained.

3. The paper presents extensive implementation details, but lacks sufficient motivation and justification for the proposed design choices. It remains unclear why the method should work as intended.

4. There is a typo in the task name: the description of Figure 3 refers to the "GWREC task," while the core task proposed in the paper is "WGREC (Weakly Supervised Generalized Referring Expression Comprehension)."

5. The evaluation strategy adopts a single threshold—has the basis for selecting this threshold been fully demonstrated? (For instance, the reason for choosing 0.7 is not explained, and the impact of different thresholds on the results is not mentioned.)

6. Although the paper conducts ablation experiments, it only compares scenarios with and without the description reconstruction and graph fusion strategies. (It fails to further analyze the interaction and impact between the visual graph, categorical graph, and description graph.)

7. The paper only focuses on performance metrics (e.g., Pr@(F1=1, IoU≥0.5)) and does not compare the computational cost (e.g., training time) between GKDN and existing methods.

8. Although a baseline has been established for the WGREC task, its performance is inferior to that of MiniGPT-v2. The reasons for this performance gap have not been analyzed in depth—for example, whether the gap stems from issues with the model structure or pre-trained data.

**Questions:**

Refer to the weaknesses above.

---

### Official Review · Reviewer_AEhY · 2025-10-29

**Soundness:** 2
**Presentation:** 1
**Contribution:** 2
**Rating:** 2
**Confidence:** 4

**Summary:**

- This work introduce a new setting, Weakly Supervised Generalized Referring Expression Comprehension (WGREC), which extends traditional WREC to handle more realistic and complex scenarios.
- To address this task, this work designs a graph-based knowledge distillation network (GKDN) guided by a large language model (LLM). This information by LLM helps the designed network build attention graphs that model the link between objects and the expression while filtering out irrelevant candidates and achived better performance.

**Strengths:**

- This work introduce a new setting, Weakly Supervised Generalized Referring Expression Comprehension (WGREC), which extends traditional WREC to handle more realistic and complex scenarios.

- this work designs a graph-based knowledge distillation network (GKDN) guided by a large language model (LLM). This information by LLM helps the designed network build attention graphs that model the link between objects and the expression while filtering out irrelevant candidates and achived better performance.

**Weaknesses:**

- The writing of this work is quite poor
  - In Table-3, the author describes that GKDN w/ G represent GKDN using Grounding DINO as teacher model, but I don't find the GKDN w/ G. Is it GKDN w/ L?
  - Line-218 describes that `the resulting region is then fed into the description construction module for description generation`. How do the generated descriptions used for network learning? This is quite confusing.
  - What does MULTI-DYNARECON means ? The authors do not give a detailed introduction.


- The technical conribution and insight is weak.
  - The work `GRES`~[A] has proposed similar setting, which decreased the originality of this work.
  - The core idea is to utilize the MLLM to generate more reliable supervision information for training the proposed graph network, which lacks technical innovation.
  -  The author claims that the proposed evaluation strategy enables their method to better adapt to the diverse scenarios in GREC. It is very strange that the evaluation strategy is be specially designed for the current method. The authors even do not provide a explanation about the evaluation metric.
  - The performance of network distilled from LLM is far behind the LLM and even some zero-shot methods (e.g., `GroundVLP` ~[B] and `LocalizationHeads` ~[C]). This makes it uncertain whether the work is actually feasible. Also, the authors do not provide any code for reproduce their work.

Overall, I find this paper to be rather rough in both writing and content insight, falling below the bar for conference acceptance.

[A] CVPR2023, GRES: Generalized Referring Expression Segmentation

[B] AAAI2024, GroundVLP: Harnessing Zero-shot Visual Grounding from Vision-Language Pre-training and Open-Vocabulary Object Detection

[C] CVPR2025, Your Large Vision-Language Model Only Needs A Few Attention Heads For Visual Grounding

**Questions:**

Refer to Weaknesses.

---

### Official Review · Reviewer_dpBj · 2025-10-30

**Soundness:** 3
**Presentation:** 2
**Contribution:** 3
**Rating:** 4
**Confidence:** 3

**Summary:**

This paper addresses the limitations of the traditional weakly supervised referential expression comprehension (WREC) task and proposes a new task: weakly supervised generalized referential expression comprehension (WGREC). The authors design a graph-based knowledge distillation network (GKDN), using a large language model (LLM) as the teacher model. The LLM generates descriptions and pseudo-target locations to assist in constructing an attention graph, thereby providing localization and comprehension capabilities. It outperforms the existing state-of-the-art techniques on the WREC task and shows potential in the WGREC task.

**Strengths:**

1.	The traditional WREC task ignores the "multi-target" and "no target" scenarios, while the WGREC task is proposed for the first time in this paper. The core goal of WREC is to locate all the targets described by the expression and return null results when there is no match, which extends the application scope of the coreference understanding task and meets the requirements of real world applications.
2.	WREC task only relies on image-expression pairs without precise object location annotations. The existing VLM model has strong detection ability, and it is very reasonable and smooth to use the pseudo-target location and description generated by it as auxiliary information.
3.	GKDN outperforms most SOTA methods on the WREC task. Ablation experiments verify the effectiveness of each part, including key variables such as LLM selection, number of inference steps, and effectiveness of graph structure.

**Weaknesses:**

1.	The description of the experimental scene is not clear, and the content of the diagram is not rigorous enough. Are the LLM models in Table 3 the experimental results under the WGREC task or WREC task? "Using a larger model such as Grounding DINO may further improve GKDN s performance." mentioned in 4.2, but not found results using Grounding DINO. Should "GKDNw/L" in Table 3 be "GKDNw/G"?
2.	Although GKDN is the first baseline for WGREC, its weakly supervised version (GKDN w/ W) performs much worse than the teacher model MiniGPT-v2 on the gRefCOCO dataset, and the core reasons for the performance gap are not analyzed.
3.	There is a lack of comparison of experimental results in WGREC task scenarios, such as with some existing large models Qwen-VL-2.5 or Qwen-VL-3. If there is a gap in accuracy, the advantages of existing methods in other aspects can be analyzed, such as the number of parameters, training time, inference speed, etc.
4.	The authors point out the "high computational cost" of traditional contrastive learning, but do not quantify the efficiency advantage of GKDN.

**Questions:**

1.	Is GroundingDINO in Table 3 a trained or untrained model?
2.	Why choose MiniGPT-v2 and GroundingDINO as teacher models? Is it possible to select some newer large models to provide more accurate pseudo-object location and description?

---

### Note · Authors · 2025-11-14

I have read and agree with the venue's withdrawal policy on behalf of myself and my co-authors.